# The association between vitamin D serum levels, supplementation, and suicide attempts and intentional self-harm

Jill E. Lavigne[1,2], Jason B. Gibbons[1,3]*

**1** Department of Veterans Affairs, Center of Excellence for Suicide Prevention, Canandaigua, New York, United States of America, **2** Wegmans School of Pharmacy, St John Fisher College, Rochester, New York, United States of America, **3** Department of Health Policy & Management, Johns Hopkins Bloomberg School of Public Health, Baltimore, MD, United States of America

* jgibbo13@jhu.edu

## Abstract

### Objectives

The purpose of this study is to determine the associations between Vitamin D supplementation, 25(OH) blood serum levels, suicide attempts, and intentional self-harm in a population of veterans in the Department of Veterans Affairs (VA).

### Methods

A retrospective cohort study of US Veterans supplemented with Vitamin D. Veterans with any Vitamin $D_3$ (cholecalciferol) or Vitamin $D_2$ (ergocalciferol) fill between 2010 and 2018 were matched 1:1 to untreated control veterans having similar demographics and medical histories. Cox proportional hazards regression was used to estimate the time from the first Vitamin $D_3$ (cholecalciferol) or Vitamin $D_2$ (ergocalciferol) prescription fill to the first suicide attempt or intentional self-harm. Analyses were repeated in stratified samples to measure associations by race (Black or White), gender (male or female), blood levels (0–19 ng/ml, 20–39 ng/ml, and 40+ ng/ml), and average daily dosage.

### Results

Vitamin D3 and D2 supplementation were associated with a 45% and 48% lower risk of suicide attempt and self-harm (($D_2$ Hazard Ratio (HR) = 0.512, [95% CI, 0.457, 0.574]; $D_3$ HR = 0.552, [95% CI, 0.511, 0.597])). Supplemented black veterans and veterans with 0–19 ng/ml vitamin D serum levels were at ~64% lower risk relative to controls (Black Veteran HR: 0.362 [95% CI: 0.298,0.440]; 0–19 ng/ml HR: 0.359 [95% CI: 0.215,0.598]). Supplementation with higher vitamin D dosages was associated with greater risk reductions than lower dosages (Log Average Dosage HR: 0.837 [95% CI: 0.779,0.900]).

**Data Availability Statement:** The data that support the findings of this study are available from the United States Department of Veterans Affairs, but restrictions apply to the availability of these data, which were used for the current study, and so are

not publicly available. Data are, however, available from the authors upon reasonable request and with permission of the United States Department of Veterans Affairs. You may contact Erika L. Trumble (erka.trumble@va.gov) to request data access information.

**Funding:** This study was supported, in part, with resources from the Department of Veterans Affairs Center of Excellence for Suicide Prevention. Support for VA/CMS data was provided by the Department of Veterans Affairs, VA Health Services Research and Development Service, VA Information Resource Center (Project Numbers SDR 02-237 and 98-004), awarded to authors JG and JL. The views expressed in this article are those of the authors and do not necessarily reflect the position or policy of the Department of Veterans Affairs or the United States government.

**Competing interests:** The authors have declared that no competing interests exist.

## Conclusions

Vitamin D supplementation was associated with a reduced risk of suicide attempt and self-harm in Veterans, especially in veterans with low blood serum levels and Black veterans.

## Introduction

Serum vitamin D is derived from skin exposure to the sun or dietary Vitamin D, including supplements $D_2$ and $D_3$. Serum vitamin D supports bone health, immune function, and absorption of other micronutrients [1]. Vitamin D insufficiency and deficiency are prevalent in the US [2]. More than 30% of US military members have been shown to have 25-hydroxyvitamin D [25(OH)] levels below 20ng/ml [3], which is considered deficient [4]. Vitamin deficiency is particularly prevalent among service members of color and males [3]. Servicemembers and veterans also have elevated suicide attempt and suicide rates [5].

A growing body of evidence has identified associations between suicidal behavior [3, 6–13] or depressive symptoms [14–21] and low levels of serum 25-hydroxyvitamin D (serum Vitamin D). In a case-control study of 495 service members who had been deployed and later died by suicide, those with seasonally-adjusted Vitamin D serum levels [(25-hydroxyvitamin D [25(OH)]) less than 15.5 ng/mL had the highest risk of a suicide attempt [3]. In a study of 157,211 healthy Korean veterans, those with 25 (OH) levels below 10ng/mL were significantly more likely to have experienced suicidal ideation [22]. Vitamin D deficiency has previously been associated with symptoms similar to depression, including fatigue, mood changes (e.g., hopelessness and sadness), suicidal thoughts, anxiety, changes in appetite and weight, insomnia, and forgetfulness [1]. In addition, low Vitamin D serum levels have been associated with other mental and physical disorders with high rates of comorbid depression, including obesity, schizophrenia, and seasonal affective disorder [1].

A potential mechanism of action for vitamin D serum levels and suicide was found in a study of the post-mortem brain tissue from 15 depressed suicide decedents matched by age, sex, and death interval to 15 non-psychiatric controls [9]. Specifically, increased vitamin D receptor gene expression and decreased cathelicidin-related antimicrobial peptide expression were identified in the suicide decedents but not the matched controls. [9] However, randomized controlled trials of vitamin D supplementation and psychiatric outcomes have produced conflicting results, with trials reporting both positive [16, 17, 19–21, 23] and null or negative effects [14, 15, 18, 24–26]. The US Preventive Services Task Force reviewed 11 trials of Vitamin D supplementation and all-cause mortality but found no differences in mortality between the intervention and control groups in any study [27]. Notably, all of these trials, including those of mortality and incident depression, included subjects whose 25(OH) levels were above the level (<20ng/mL) defined by the National Academies of Medicine as deficient. Due to insufficient evidence [28–30], the US Preventive Services Task Force (USPSTF) was unable to make a recommendation for or against Vitamin D supplementation in the general adult population in 2021 [2]. The USPSTF has since called for research to more clearly define serum level cut-offs for insufficiency and deficiency as well as associations between vitamin D blood serum levels and diseases, including psychiatric disorders [2, 27, 29].

This study aimed to explore associations between 25(OH) levels, vitamin $D_3$ or $D_2$ supplementation, and suicide attempts or self-harm among U.S. veterans. We also evaluated the moderation of these associations by veteran race, vitamin D serum level, dosage, and gender.

## Materials & methods

### Study design

This retrospective cohort study included Department of Veterans Health Affairs (VHA) veterans represented in the national VHA Corporate Data Warehouse (CDW) with at least one electronic medical or pharmacy record between 2010–2018 linked at the individual level to Medicare inpatient, outpatient, professional (i.e., carrier), beneficiary summary (i.e., MBSF), and Part D (Drug) files for the same period. All veterans who received Vitamin $D_2$ (ergocalciferol) or Vitamin $D_3$ (cholecalciferol) fills between 2010 and 2018 were separately identified and matched 1:1 to untreated veterans on their propensity for supplementation (Table 1). After matching, Cox proportional hazards models were used to estimate the association between supplementation with Vitamin $D_2$ or $D_3$ (ergocalciferol or cholecalciferol) and suicide attempts. Veterans were followed from their index prescription for Vitamin D and censored at the first instance of a suicide attempt or self-harm, death, 24 months, or the end of the study period in December 2018. Analyses were repeated using stratified patient samples to see if patient race, gender, vitamin D dosage, and vitamin D blood serum levels modified the association between supplementation and suicide attempts and intentional self-harm.

### Study population

Data from the Department of Veterans Affairs Corporate Data Warehouse (CDW), including inpatient, outpatient, laboratory, prescription fill, and demographic data, were used to identify cohorts of veterans treated with and without Vitamin D supplementation in 2010–2018. Data were linked to Medicare claims at the individual patient level inside the VHA secure computing environment. Veterans with one or more filled prescriptions or medical claims between 2010–2018 were identified as having filled prescriptions for vitamin $D_2$, vitamin $D_3$, or neither (Table 1). We excluded veterans who filled both vitamin $D_2$ and $D_3$ and those with less than 90 days of supplementation in the two years following their index (i.e., first) vitamin D prescription. We also excluded veterans who had one or more prescriptions for vitamin $D_2$ between 2010–2018 from the vitamin $D_3$ control group and veterans who had one or more prescriptions for vitamin $D_3$ between 2010–2018 from the vitamin $D_2$ control group. This was done to prevent Vitamin $D_3$ and $D_2$ treated veterans from being used as controls for the $D_2$ and $D_3$ analyses. We used 1-to-1 propensity score matching from the remaining sample to identify a cohort of 169,241 vitamin $D_2$-treated veterans and 490,885 vitamin $D_3$-treated veterans, each matched to an equal number of controls (Table 1). Covariates (Table 1) used to generate propensity scores included common indications for vitamin D (e.g., vitamin D deficiency; see Table 1 for a complete list of conditions), demographic (i.e., age, race/ethnicity, and gender), and mental health comorbidities previously associated with suicide attempts in the academic literature (i.e., major depressive disorder, mood disorder, schizophrenia, bipolar disorder, substance use disorder, post-traumatic stress disorder, and personality disorder). Health conditions were identified using the presence of one or more ICD-9 or ICD-10 diagnoses on any medical claim for a veteran between 2010–2018. Propensity score matching was performed to help balance the treated and control populations across characteristics associated with vitamin D supplementation and suicidal and self-harm behaviors.

We also created subgroups for use in stratified analyses (Table 2). Subgroups were sampled from the initial vitamin $D_2$ and $D_3$ cohorts separately to determine subgroup heterogeneity in association with suicide attempts and self-harm rates by vitamin $D_2$ versus $D_3$. Subgroups included cohorts of individuals by race (black versus white), gender (male versus female), and vitamin D blood serum levels (i.e., 0–19 ng/ml, 20–39 ng/ml, and $\geq 40$ ng/ml). Vitamin D blood serum levels were identified by the last 25-hydroxyvitamin D or 1,25-dihydroxyvitamin

**Table 1. Characteristics of veterans receiving vitamin D and controls before and after 1-1- propensity score matching.**

| Covariate | Vitamin $D_2$ | | | | Vitamin $D_3$ | | | |
|---|---|---|---|---|---|---|---|---|
| | Pre-Matching | | Post-Matching Restricted Sample | | Pre-Matching | | Post-Matching Restricted Sample | |
| | Control | Treated | Control | Treated | Control | Treated | Control | Treated |
| **N** | 1,764,309 | 176,403 | 169,241 | 169,241 | 1,907,720 | 538,442 | 490,885 | 490,885 |
| **Age** | 62 | 60 | 60 | 60 | 62 | 60 | 61 | 61 |
| **Gender** | | | | | | | | |
| Female | 8.4% | 13.7% | 13.3% | 13.4% | 8.6% | 12.2% | 11.5% | 11.7% |
| Male | 91.6% | 86.3% | 86.7% | 86.6% | 91.4% | 87.8% | 88.5% | 88.3% |
| **Race/Ethnicity** | | | | | | | | |
| African American | 14.2% | 27.8% | 26.8% | 27.2% | 14.3% | 21.1% | 20.0% | 20.1% |
| Asian | 1.7% | 0.9% | 0.9% | 0.9% | 1.3% | 1.6% | 1.5% | 1.6% |
| Native American | 1.8% | 1.7% | 1.7% | 1.7% | 1.8% | 2.3% | 2.2% | 2.2% |
| Other Race | 4.0% | 4.0% | 4.2% | 4.1% | 4.1% | 4.8% | 4.8% | 4.8% |
| White | 78.4% | 65.6% | 66.4% | 66.1% | 78.5% | 70.2% | 71.4% | 71.4% |
| **Mental Health Condition** | | | | | | | | |
| Bipolar | 2.9% | 3.7% | 4.0% | 3.7% | 3.0% | 5.4% | 5.2% | 5.0% |
| Depression | 27.0% | 35.8% | 36.3% | 35.6% | 28.2% | 42.8% | 41.9% | 40.9% |
| Mood disorder | 20.1% | 28.1% | 28.5% | 27.8% | 21.2% | 34.8% | 33.6% | 32.8% |
| Personality Disorder | 1.9% | 2.6% | 2.8% | 2.6% | 2.0% | 3.6% | 3.0% | 2.9% |
| PTSD | 17.3% | 22.9% | 23.5% | 22.6% | 18.5% | 31.7% | 30.1% | 29.4% |
| Schizophrenia | 0.9% | 1.1% | 1.3% | 1.1% | 1.0% | 1.7% | 1.7% | 1.6% |
| Sleep Disorder | 4.2% | 5.2% | 5.2% | 5.1% | 4.4% | 6.7% | 6.6% | 6.4% |
| Substance Use Disorder | 24.6% | 33.7% | 34.7% | 33.6% | 26.1% | 35.6% | 35.9% | 34.7% |
| **Physical Condition** | | | | | | | | |
| Anemia | 19.9% | 21.0% | 21.0% | 20.9% | 19.7% | 22.0% | 22.0% | 21.9% |
| Congestive Heart Failure | 11.0% | 10.2% | 9.7% | 10.2% | 10.7% | 10.2% | 9.9% | 10.3% |
| Diabetes | 25.2% | 30.8% | 31.4% | 30.8% | 25.3% | 31.9% | 32.4% | 31.5% |
| Fatigue | 21.9% | 21.1% | 21.3% | 21.0% | 21.6% | 22.7% | 23.3% | 22.7% |
| Hypercholesterolemia | 15.1% | 13.9% | 14.6% | 13.8% | 14.9% | 15.0% | 16.0% | 15.1% |
| Hyperlipidemia | 56.6% | 63.0% | 64.9% | 63.0% | 57.0% | 66.0% | 67.7% | 65.5% |
| Hypertension | 58.8% | 65.4% | 66.4% | 65.4% | 58.9% | 66.1% | 67.2% | 65.7% |
| Hypothyroidism | 12.2% | 11.6% | 16.4% | 11.6% | 12.2% | 13.1% | 13.5% | 13.2% |
| Limb Pain | 17.7% | 20.4% | 21.8% | 20.3% | 17.5% | 21.4% | 22.3% | 20.9% |
| Long Term use of Medication | 30.6% | 33.0% | 34.2% | 32.9% | 30.6% | 35.3% | 36.3% | 34.9% |
| Needs Flu Shot | 64.5% | 71.8% | 73.4% | 71.7% | 65.0% | 75.0% | 76.6% | 74.5% |
| Reflux | 33.6% | 36.9% | 38.0% | 36.8% | 33.8% | 41.2% | 41.9% | 40.5% |
| Urinary Tract Infection | 15.4% | 16.0% | 16.1% | 15.8% | 15.2% | 16.9% | 17.0% | 16.8% |
| Vitamin D Deficiency | 21.4% | 54.0% | 53.8% | 54.3% | 22.9% | 52.4% | 48.5% | 50.2% |

Note: Treated beneficiaries include all beneficiaries that received one or more vitamin D prescriptions between January 1, 2010, and December 31, 2018. Conditions were measured based on the presence of one or more ICD-9 or ICD-10 codes during the entire study period.

Source: VA and Medicare Claims Data.

D test before the index prescription of a vitamin D supplement for a patient in either the D2 or D3 treatment group or the corresponding treated pair's index prescription for the untreated controls. Therefore, vitamin D blood serum levels reflect pre-treatment (or baseline) values. We provide counts of treated and control veterans in each subgroup in Table 2.

**Table 2. Frequencies of suicide attempts and intentional self-harm by veteran cohort.**

| Veteran Cohort | Total Veterans N | Control N | Treated N | Suicide Attempt and Intentional Self-Harm by Control N (%) | Suicide Attempt and Intentional Self-Harm by Treated N (%) |
|---|---|---|---|---|---|
| **Vitamin $D_2$** | | | | | |
| **All Veterans** | 338,482 | 169,241 | 169,241 | 878 (0.52%) | 452 (0.27%) |
| **Gender** | | | | | |
| *Male* | 293,205 | 146,704 | 146,501 | 773 (0.50%) | 376 (0.26%) |
| *Female* | 45,277 | 22,537 | 22,740 | 145 (0.64%) | 76 (0.33%) |
| **Race** | | | | | |
| *Black* | 91,446 | 45,362 | 46,084 | 243 (0.54%) | 105 (0.23%) |
| *White* | 211,655 | 104,724 | 106,931 | 553 (0.53%) | 305 (0.29%) |
| **0–19 Vitamin D Level Cohort** | 57,681 | 3,800 | 53,881 | 6 (0.16%) | 77 (0.14%) |
| **20–39 Vitamin D Level Cohort** | 37,356 | 14,335 | 23,021 | 20 (0.14%) | 34 (0.15%) |
| **40+ Vitamin D Level Cohort** | 5,377 | 4,591 | 786 | 5 (0.11%) | 1 (0.13%) |
| **Vitamin $D_3$** | | | | | |
| **All Veterans** | 981,770 | 490,885 | 490,885 | 1,786 (0.36%) | 991 (0.20%) |
| **Gender** | | | | | |
| *Male* | 867,800 | 433,450 | 434,350 | 1,499 (0.35%) | 833 (0.19%) |
| *Female* | 113,970 | 56,535 | 57,435 | 287 (0.51%) | 158 (0.28%) |
| **Race** | | | | | |
| *Black* | 196,732 | 98,164 | 98,568 | 383 (0.39%) | 138 (0.14%) |
| *White* | 666,322 | 329,024 | 337,298 | 1,209 (0.37%) | 766 (0.23%) |
| **0–19 Vitamin D Level Cohort** | 81,194 | 9,534 | 71,660 | 20 (0.21%) | 55 (0.08%) |
| **20–39 Vitamin D Level Cohort** | 153,887 | 41,678 | 112,209 | 51 (0.12%) | 116 (0.10%) |
| **40+ Vitamin D Level Cohort** | 20,940 | 14,327 | 6,613 | 5 (0.03%) | 6 (0.09%) |

## Exposure

The primary exposure was supplementation with vitamin $D_3$ (cholecalciferol) or vitamin $D_2$ (ergocalciferol), including multivitamins. Specifically, the association between suicide attempts and self-harm and supplementation was captured by the parameter estimates on an indicator variable equal to one (supplementation) or zero (control). We used the natural logarithm of average vitamin D dosage as a secondary exposure. The average vitamin D dosage was obtained by taking the average of the prescribed or administered dosages (which ranged from 40 UI to 50,000 UI) weighted by days supplied during the observation period during which the patient filled prescriptions for vitamin D (i.e., two years following index prescription). For veterans not taking vitamin D, we set the average daily dosage and its natural logarithm to be 0 to prevent constructing undefined values. Using the natural logarithm allowed us to relate percentage-point increases in average daily dosage to the associated probability of suicide attempt or self-harm in our statistical model. Single-unit increases in vitamin D dosage (i.e., 1 ng/ml) are unlikely to be related to changes in suicide and self-harm risk, which motivated our use of the natural logarithm of the average daily dosage. Vitamin $D_3$ and $D_2$ prescriptions and dosages were identified using National Drug Codes in the Department of Veterans Affairs Electronic Health Records data and the Medicare Part D claims data (i.e., Medicare prescription drug claims file).

## Outcomes

The primary outcome identified in VHA and Medicare data was suicide attempt or intentional self-harm resulting in an emergency department (ED) or inpatient admission (from January 1, 2010) as identified by any mention at the first record (initial, subsequent or sequelae) of ICD–10–CM codes: X71–X83, T36–T50 with the 6th character of 2 (except for T36.9, T37.9, T39.9, T41.4, T42.7, T43.9, T45.9, T47.9, and T49.9, which are included if the 5th character is 2), T51–T65 with the 6th character of 2 (except for T51.9, T52.9, T53.9, T54.9, T56.9, T57.9, T58.0, T58.1, T58.9, T59.9, T60.9, T61.0, T61.1, T61.9, T62.9, T63.9, T64.0, T64.8, and T65.9, which are included if the 5th character is 2), T71 with the 6th character of 2, and T14.91.) About 90% of all injury ED visits and hospitalizations were assigned an external cause of injury code by 2013.

## Statistical analysis

We used a Cox proportional hazards model to compare vitamin $D_2$ and $D_3$ supplementation groups to matched controls on time to first suicide attempt or intentional self-harm. The only covariate in the model was an indicator representing vitamin D supplementation (i.e., "1" if supplemented and "0" if untreated control). The model was run separately for vitamin $D_2$ and $D_3$ treated veterans and their matched controls. Associations were estimated separately due to potential differences in outcomes between Vitamin $D_2$ and $D_3$ due to differences in metabolism. In both models, veterans were followed from their index prescription fill (or, for the matched treated pair's control, the index prescription date) and were censored at the first instance of suicide attempt or self-harm, death, after two years, or the end of the study period in December 2018, whichever came first. The model was then estimated separately for each of the subgroups, which included race (black versus white), gender (female versus male), and vitamin D blood serum level (0–19 ng/ml, 20–39 ng/ml, and 40+ ng/ml). For the Vitamin D blood serum level subgroup analyses, we also included specifications that modeled the association between the natural logarithm of the average daily dosage across the entire treatment episode and suicide attempts and intentional self-harm.

This study (VA MIRB # 00701, PI Jill Lavigne) was reviewed and approved under Category 4 exempt determination by the Syracuse VA Medical Center Institutional Review Board in Syracuse, New York, and the VA's VIREC Office for Medicare data. This research did not meet the criteria for human subjects research because the VA Corporate Warehouse Data and Medicare claims are de-identified, so informed consent was not required. All methods were performed in accordance with the relevant guidelines and regulations.

## Results

Following matching, the vitamin $D_2$ and $D_3$ supplemented and control groups were similar across potential confounders (see Table 1). Moreover, most treated veterans in the $D_2$ and $D_3$ cohorts were successfully matched to an untreated control.

Patient frequencies by analytic cohort and suicide attempts by supplemented versus control veterans are presented in Table 2. In our entire Vitamin $D_2$ sample, we identified a total of 338,482 veterans, 169,241 treated and 169,241 controls, with a 0.52% unadjusted suicide attempt and self-harm rate in controls and a 0.27% rate amongst the treated. In our entire Vitamin $D_3$ sample, we identified 981,770 veterans; 490,885 treated and 490,885 controls. The unadjusted suicide attempt and self-harm rate was 0.36% in the control population and 0.20% in the treated. Attempt and self-harm rates by gender and race were similar to the entire sample for both Vitamin $D_2$ and $D_3$ subsamples. However, the serum level subsamples in both Vitamin $D_2$ and Vitamin $D_3$ demonstrated notably lower attempt and self-harm rates than the

full sample. It may therefore be the case that those who have their vitamin D levels tested are characteristically different from those that did not receive testing. Finally, as vitamin D levels increased, the proportion of veterans experiencing suicide attempts and self-harm declined in the control groups for Vitamin $D_2$ and Vitamin $D_3$ and stayed relatively fixed in the treated groups.

## Suicide attempts and intentional self-harm in the total sample

Estimates from the Cox proportional hazards models of the association between vitamin $D_2$ and $D_3$ supplementation and suicide attempts and self-harm appear in Table 3. Vitamin $D_2$ supplementation was associated with a 48.8% reduction in suicide attempt risk, and vitamin $D_3$ supplementation with a 44.8% reduction in suicide attempt and self-harm risk ($D_2$ Hazard Ratio (HR) = 0.512, [95% CI, 0.457, 0.574]; $D_3$ HR = 0.552, [95% CI, 0.511, 0.597]). Kaplan Meier curves are displayed in Figs 1 & 2.

## Subgroup analyses

**Gender.** The association between supplementation and suicide attempts and self-harm was similar by gender for both vitamin $D_2$ and $D_3$ and was similar to the full sample estimates. Among veterans supplemented with vitamin $D_2$ (ergocalciferol), males had a 48.9% lower risk and females a 48.3% lower risk of suicide attempt (males HR = 0.511, [95% CI, 0.451, 0.579]; females HR = 0.517, [95% CI, 0.392, 0.683]). Similarly, among veterans supplemented with Vitamin $D_3$ (cholecalciferol), males were at 44.6% lower risk relative to 45.9% for females (males HR = 0.554, [95% CI, 0.509, 0.603]; females HR = 0.541, [95% CI, 0.446, 0.657]).

**Race.** The association between supplementation and suicide attempt and self-harm differed notably between Black and White veterans. Vitamin $D_2$ supplementation was associated with a 57.9% reduction in suicide attempt and self-harm risk for Black veterans compared to a 46.3% lower risk for White veterans (White veterans HR = 0.537, [95% CI, 0.467, 0.618]; Black veterans HR = 0.421, [95% CI, 0.335, 0.530]). Similarly, among veterans supplemented with vitamin $D_3$ (cholecalciferol), White veterans were at 38.7% lower risk relative to 63.8% lower risk for Black veterans (White veterans HR = 0.613, [95% CI, 0.561, 0.672]; Black veterans HR = 0.362, [95% CI, 0.298, 0.440]).

**Vitamin D serum levels.** The association between supplementation and suicide attempts and self-harm also differed by vitamin D blood serum level. In the 0–19 ng/ml blood serum cohort, Vitamin $D_3$ (cholecalciferol) supplementation was associated with a 64.1% reduction in risk relative to untreated controls (0–19 ng/ml HR = 0.359, [95% CI, 0.215, 0.598]). In this cohort of vitamin D deficient veterans, each additional percentage point increase in average daily dosage was associated with a 13.8% reduction in risk (0–19 ng/ml log average daily dosage HR = 0.837, [95% CI, 0.779, 0.900]). For veterans with blood levels between 20–39 ng/ml, while the overall association was not significant, Vitamin $D_3$ (cholecalciferol) supplementation was associated with a significant 9.6% reduction in suicide attempt and self-harm risk for each additional percentage point increase in average daily dosage (20–39 ng/ml log average daily dosage HR = 0.904, [95% CI, 0.862, 0.948]). No significant overall or dose-response associations were found for the 40+ ng/ml group. All results for Vitamin $D_2$ (ergocalciferol) by dosage and blood level were statistically insignificant.

## Discussion

Our findings suggest that supplementation with vitamin $D_2$ (ergocalciferol) or vitamin $D_3$ (cholecalciferol) is associated with a 45%-48% reduced risk of suicide attempts and self-harm, on average, among veterans in the Department of Veterans Health Affairs. Subgroup analyses

**Table 3. Estimated hazard ratios from a survival analysis comparing time-to-first-suicide-attempt or intentional self-harm between vitamin D supplemented veterans and matched control veterans.**

| Patient Cohort | Vitamin $D_2$ Treated versus Untreated Control (Hazard Ratio w/ 95% CI) | Vitamin $D_3$ Treated versus Untreated Control (Hazard Ratio w/ 95% CI) |
|---|---|---|
| **All Veterans** | 0.512*** | 0.552*** |
| | (0.457, 0.574) | (0.511, 0.597) |
| **Gender** | | |
| *Male* | 0.511*** | 0.554*** |
| | (0.451, 0.579) | (0.509, 0.603) |
| *Female* | 0.517*** | 0.541*** |
| | (0.392, 0.683) | (0.446, 0.657) |
| **Race** | | |
| *Black* | 0.421*** | 0.362*** |
| | (0.335, 0.530) | (0.298, 0.440) |
| *White* | 0.537*** | 0.613*** |
| | (0.467, 0.618) | (0.561, 0.672) |
| **0–19 Vitamin D Level Cohort** | | |
| *All Dosages* | 0.887 | 0.359*** |
| | (0.386, 2.034) | (0.215, 0.598) |
| *Log Average Dosage* | 0.955 | 0.837*** |
| | (0.911, 1.000) | (0.779, 0.900) |
| **20–39 Vitamin D Level Cohort** | | |
| *All Dosages* | 1.100 | 0.832 |
| | (0.628, 1.929) | (0.598, 1.15) |
| *Log Average Dosage* | 0.957 | 0.904*** |
| | (0.893, 1.026) | (0.862, 0.948) |
| **40+ Vitamin D Level Cohort** | | |
| *All Dosages* | 1.151 | 2.54 |
| | (0.135, 9.856) | (0.778, 8.35) |
| *Log Average Dosage* | 0.971 | 1.045 |
| | (0.805, 1.170) | (0.867, 1.260) |

Notes: Parameters expressed, except for cases where log average dosage are referenced, are for an indicator variable set to 1 if treated and 0 if control. Average dosage is measured as the logarithm of the weighted average prescription dosage (weighted by days supplied) during the patient follow-up period (i.e., two years following the index prescription). Blood levels were based on the last patient lab value preceding the index prescription date for the treated veterans and the matched treated pair's index prescription date for the untreated controls. All Dosages analyses relate a supplementation indicator variable to suicide attempts and self-harm while the log-average dosages analyses relate the logarithm of the weighted average prescription dosage measure to suicide attempts and self-harm.

\* P < .05

\*\* P < .01

\*\*\* P < .001

Source: Veterans Affairs Chronic Data Warehouse Electronic Medical Records data and Medicare Claims Data.

demonstrated the strongest associations among Black veterans and those with vitamin D deficiency (0–19 ng/ml serum levels). Only vitamin $D_3$ (cholecalciferol) demonstrated a dose-response effect in deficient veterans, with more significant reductions in suicide attempt and self-harm risk at higher doses of vitamin $D_3$ (cholecalciferol) supplementation. This finding is

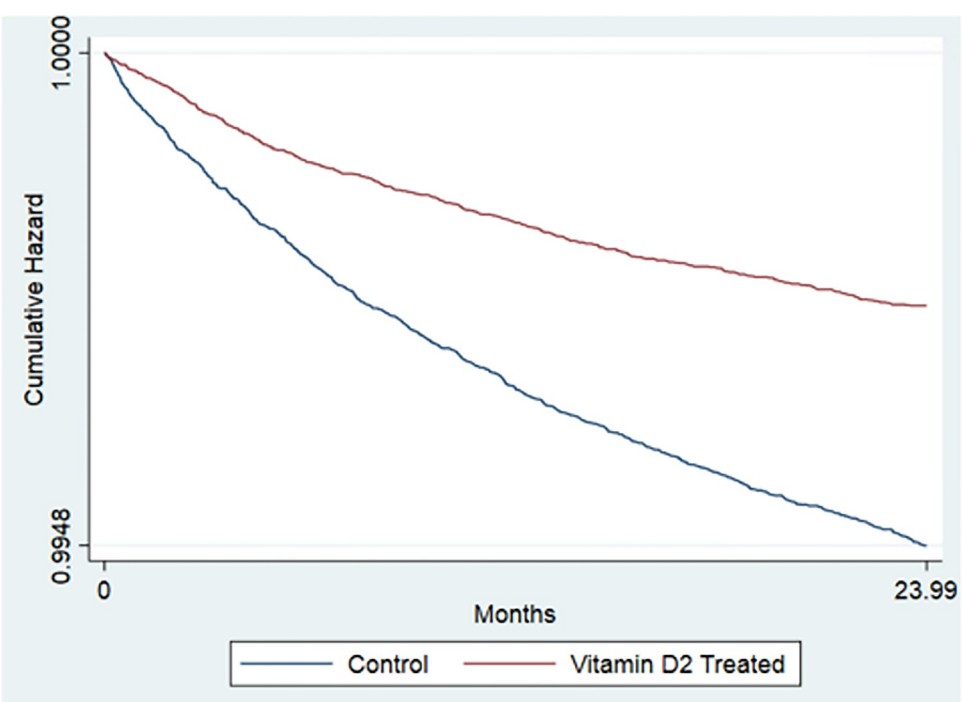

**Fig 1. Cumulative survival curves comparing vitamin D₂ treated against matched controls on time to first suicide attempt or intentional self harm.** Source: Veterans Affairs Chronic Data Warehouse Electronic Medical Records data and Medicare Claims Data.

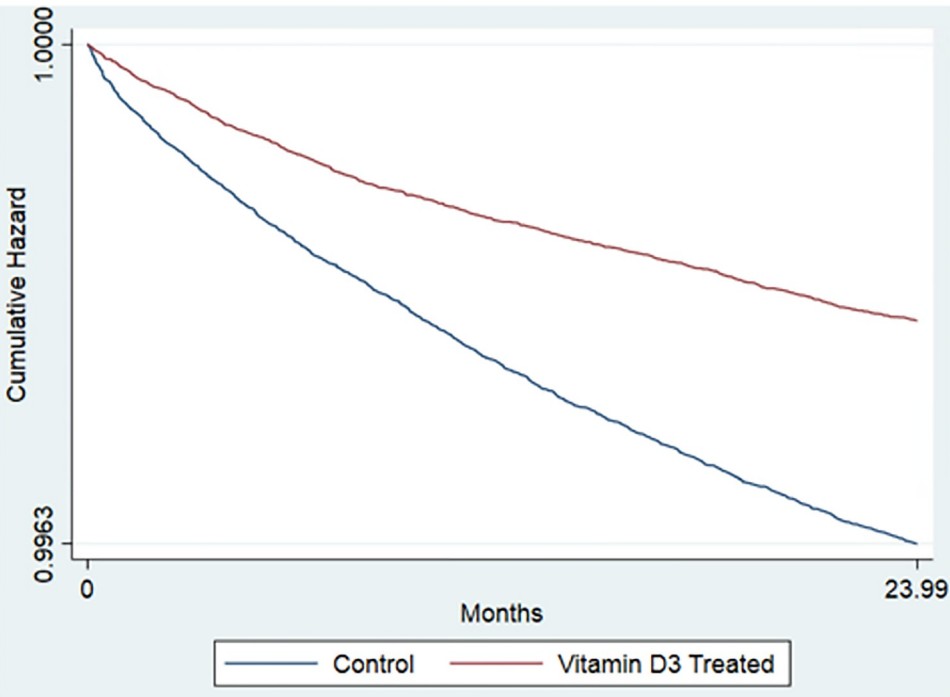

**Fig 2. Cumulative survival curves comparing vitamin D₃ treated against matched controls on time to first suicide attempt or intentional self harm.** Source: Veterans Affairs Chronic Data Warehouse Electronic Medical Records data and Medicare Claims Data.

consistent with the pharmacology of the two D vitamins. When supplements are comprised of plant-based ingredients ($D_2$, ergocalciferol) rather than from animal-based ingredients ($D_3$, cholecalciferol), the inactive vitamin D storage form (25-hydroxyvitamin D2) has a shorter half-life. Cholecalciferol (D3) has been shown to increase serum 25(OH)D more efficiently than ergocalciferol (D2). Higher doses of vitamin D are more likely to achieve and maintain serum levels than low-dose prescriptions, which may explain the association between treatment intensity and suicide attempts and self-harm [31]. Moreover, those with the lowest blood levels may benefit the most from supplementation. The dose-response finding combined with more significant risk reductions in veterans with the lowest blood serum levels adds confidence to our primary finding of a general decrease in suicide attempt and self-harm risk for veterans receiving Vitamin D supplementation. However, additional research is required to determine the association between suicidal and self-harm behavior risk once sufficient vitamin D levels are obtained.

Low vitamin D serum levels may be prevalent among veterans and service members. An analysis of the Department of Defense Serum Repository found that more than 30% of subjects had 25(OH)D values below 20ng/ml [8]. No differences in mean 25(OH)D serum levels were found between those who died by suicide and those who did not, but those in the lowest season-adjusted octile had the highest risk of suicide. A secondary analysis of these data suggested the need for further research on vitamin D, including precision medicine approaches specifically for suicide prevention [32]. This need for precision medicine (i.e., focusing Vitamin D supplementation on those with deficient levels) approaches is further supported by twelve clinical trials of vitamin D supplementation and depression reported between January 2010 and May 2020 with overall negative results. Heterogenous designs may have contributed to divergent findings. The largest trial (Bertone-Johnson 2012) of 36,282 women did not include 25(OH)D levels and found no effect of vitamin D supplementation and depression after two years [26]. Similarly, a trial of 18,535 subjects aged 50 or older with a relatively high mean baseline 25(OH) level of 30.8ng/mL randomized to placebo or 2,000 IU/d of cholecalciferol (Vitamin D3) plus fish oil found no differences in the incidence of depression or mood scores after a mean of 5.3 years of follow-up [33]. However, suicidal ideation and behavior were not assessed. Two smaller trials with shorter follow-up periods found no effect of vitamin D supplementation versus placebo on depression measures [16, 34]. Other trials included subjects with relatively normal or high 25(OH)D levels [14–21]. Supplementation dosing ranged widely from daily doses of 400 IU to 4,800 IU over eight weeks to 12 months, weekly doses of 40,000 IU to 50,000 IU over eight weeks to six months, and annual doses of 50,000 IU over three to five years. Depression screening and assessment tools included 17 instruments, such as the PHQ-9, HMA-A, WHO-5, HAM-D17, and others.

## Practice and policy implications

The Department of Veterans Health Affairs offers 25(OH)D testing to veterans, but the USPSTF does not support screening in asymptomatic adults as of 2021. Preventive services recommended by the USPSTF are required to be covered without patient cost-sharing by health plans under the Patient Protection and Affordable Care Act (2010). Although the USPSTF called for further research on Vitamin D screening in asymptomatic adults, screening may not be readily available to most US adults. However, screening may be indicated for veterans exhibiting warning signs of suicide or those with a history of suicidal behavior or ideation, particularly veterans whose suicide rates and vitamin D deficiency rates have been demonstrated to be higher than others.

The U.S. Recommended Dietary Allowance (RDA) for most adults is 600 IU of vitamin D per day. Pending confirmatory randomized controlled trials, providers may wish to initiate low-dose vitamin D supplementation, for example, at the US RDA level of 600 IU per day, without screening in patients with a history of suicidal behavior or ideation or who exhibit warning signs of suicidal behavior. Emerging evidence suggests a possible correlation between low vitamin D levels and depression [4]. Vitamin D receptors are located in areas of the brain involved in developing depression, including the hippocampus and hypothalamus [5, 6]. Adjuvant treatment of depression with vitamin D supplementation has been recommended [9]. Risk of toxicity can be managed by monitoring as toxicity is typically symptomatic, including nausea and gastrointestinal symptoms, before proceeding to urinary tract stones. However, toxicity has been shown to occur only at doses above 60,000 IU daily over several weeks, ten times the USRDA.

When recommending vitamin D supplementation, providers may wish to describe the potentially greater effectiveness of $D_3$, particularly because $D_3$ supplements are typically more expensive than $D_2$.

## Limitations

Several limitations may affect the internal validity and generalizability of our study. While many disabled veterans and those with very high medication costs receive free care from the VA, including non-prescription products, for many veterans, the cost of vitamin D products may be the same or lower when purchased through retail outlets. Therefore, we suspect some of our untreated sample, especially those with vitamin D deficiency, received over-the-counter supplementation. However, unobserved supplementation in the control group would make our estimates of decreased suicide attempt risk conservative. Another limitation is that patients filling prescriptions for Vitamin D may engage in other health and mental health improving behaviors that we do not observe. Further, although we attempt to control for some characteristics likely to be associated with vitamin D supplementation and/or suicidal and self-harm behaviors, there are many unobservable characteristics (for example, traumatic brain injury common among veterans) that could confound our results. Our findings should therefore be interpreted as associations rather than causal effects. Concerning generalizability, the VA is primarily male and middle-aged, so our associations may not be generalizable to other populations.

## Conclusion

Oral vitamin D is associated with a suicide attempt and intentional self-harm risk reduction of approximately 45%-48%. Supplementation with higher daily dosages of vitamin $D_3$ was associated with lower suicide attempt and self-harm risk than supplementation with lower dosages. Further, the associated risk reduction in suicide attempt and self-harm was more significant among Black veterans receiving supplementation with Vitamin D than white veterans, among whom low Vitamin D serum levels are more common than among White veterans. As a relatively safe, easily accessible, and affordable medication, supplementation with vitamin D in the VA may hold promise if confirmed in clinical trials to prevent suicide attempts and suicide.

## Author Contributions

**Conceptualization:** Jill E. Lavigne, Jason B. Gibbons.

**Formal analysis:** Jason B. Gibbons.

**Funding acquisition:** Jill E. Lavigne.

**Investigation:** Jill E. Lavigne.

**Methodology:** Jason B. Gibbons.

**Project administration:** Jill E. Lavigne.

**Resources:** Jill E. Lavigne.

**Software:** Jill E. Lavigne.

**Supervision:** Jill E. Lavigne.

**Validation:** Jason B. Gibbons.

**Writing – original draft:** Jill E. Lavigne, Jason B. Gibbons.

**Writing – review & editing:** Jill E. Lavigne, Jason B. Gibbons.

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
