## [Decision Letter · Decision Letter 0]

20 Sep 2022

PONE-D-22-20954

The Association between Vitamin D Serum Levels, Supplementation, and Suicide Attempts or Intentional Self Harm

PLOS ONE

Dear Dr. Gibbons,

Thank you for submitting your manuscript to PLOS ONE. After careful consideration, we feel that it has merit but does not fully meet PLOS ONE’s publication criteria as it currently stands. Therefore, we invite you to submit a revised version of the manuscript that addresses the points raised during the review process.  

We look forward to receiving your revised manuscript.

Kind regards,

James D. Clelland, Ph.D.

Academic Editor

PLOS ONE

Journal Requirements:

    "This study was supported, in part, with resources from the Department of Veterans Affairs Center of Excellence for Suicide Prevention in support of the 2020 call by the Department of Veterans Affairs for research to support the national response to the COVID-19 pandemic.  This study does not represent the views of the Department of Veterans Affairs or the United States Government "

   "This study was supported, in part, with resources from the Department of Veterans Affairs Center of Excellence for Suicide Prevention in support of the 2020 call by the Department of Veterans Affairs for research to support the national response to the COVID-19 pandemic. This study does not represent the views of the Department of Veterans Affairs or the United States Government. Dr. Gibbons had full access to all the data in the study and takes responsibility for the integrity of the data and the accuracy of the data analysis."

 "This study was supported, in part, with resources from the Department of Veterans Affairs Center of Excellence for Suicide Prevention in support of the 2020 call by the Department of Veterans Affairs for research to support the national response to the COVID-19 pandemic.  This study does not represent the views of the Department of Veterans Affairs or the United States Government.  "

6. Please ensure that you include a title page within your main document. You should list all authors and all affiliations as per our author instructions and clearly indicate the corresponding author.

Additional Editor Comments:

The authors should address the questions and comments of both reviewers.

Please address the questions and comments of both reviewers:  Reviewer 1. " I was very glad to see the review of these studies in this paper revealing their inadequacies and emphasizing that suicidal ideation and behavior were not assessed. But I wonder if that review might be more appropriate in the introduction rather than the discussion.

This study looks at suicide attempts and intentional self-harm resulting in emergency room or in-patient admission, which are much more definite and specific than depression and have some different brain mechanisms. It is a retrospective study using Veterans Affairs and Medicare electronic data, from 2010 to 2018, with patients prescribed Vitamin D matched 1:1 to untreated controls. It uses a very large sample: over half a million treated, nearly two million not treated controls: or is it three and a half million not treated? (D2 Controls plus D3 Controls.) Why are the numbers of Pre-matching D3 controls not the same as Pre-matching D2 controls? If they are a different pool of controls how were the controls chosen for each pool? Some further explanation might be useful. And related: why are there only 9,534 D3 controls with 0-19 Vitamin D level when there are 53,881 D2 controls with that level?

The treated patients were included from the date of their prescription. Were the untreated matched by that starting date?

It is fortuitous (and surprising to me) that so many had Vitamin D blood levels, especially as many with low levels were not treated (see below)

The dose used is mentioned and analyzed using the natural logarithm of average dosage (why?). I think mention of the range of dose used would be useful – especially as there is discussion of the appropriate dose under Practical and Policy Implications – but not based on the results of this study.

I don’t understand Propensity Score matching despite reading several descriptions on line. I will accept that it is legitimate. Someone experienced with it should decide its appropriateness. Is everyone expected to know what it means? If not, some more explanation might help. For example, if I look at Depression can I presume that for each Treated person there is a Control of the same gender, race, age (how close?) and any other characteristic?

Typo Table 1, Post-Matching Restricted Sample Control column, Age ?61 not 6.

Table 1 does confirm that people with Bipolar, Depression and other psychiatric conditions were more likely to be given Vitamin D although a greater number with these conditions were not treated, but the matching brought the numbers and the treatment likelihood to the same. The same is true for Vitamin D deficiency. All good.

Table 2, typo: Suicide Attempt by Control column, 0-19 Vitamin D Level Cohort line, should be 86, not 6. And some of the numbers don’t add up; for example, Vitamin D2 females: if there are 45,277 total females the treated should be about 22,742 not 227,420. It does look as if, when Vitamin D levels were very low, D3 was used preferentially for treatment but still about 63,000 with levels 0-19 were untreated. That raises the question of why blood levels were taken.

The Vitamin D2 Level Cohorts Suicide Attempts are surprising (not much difference) given the big differences in race and gender cohorts. That makes it hard to understand the last sentence of the Results section: “However, models of the serum level subsamples demonstrated notably lower rates with increasing blood serum levels relative to the full sample.” Is that for D2 as well as D3 or just the latter? And is there a higher rate of suicide attempts with Vitamin D when the blood level was over 40. (Yes, the numbers are too small. Still it reminds me of the association between maternal (or cord) blood levels and schizophrenia in the offspring – a J-shaped curve.)

Presumably the mental health comorbidities were taken from any time during the two years of treatment (or control). It is interesting to me that the beneficial effects of treatment continue with little change for the two years (Figures 1 and 2) when blood levels of Vitamin D normalize with treatment in two to three months.

Under Discussion, second paragraph, line 4, typo: did should be died. In the sixth line the word “by” presumably should not be there. I think it would be useful to say exactly what is meant by “precision medicine” – presumably only treating those with low Vitamin D levels."

Reviewer 2.  "Methods:  It seems from Table 1 that the pre- and post-matched control groups for Vitamin D2 and D3 were independent of each other (as the starting numbers are different). Statistically speaking, this is a benefit and should be noted in the methods.

Line 86. The authors should specify the vitamin D assays employed for blood measures (e.g. 25-hydroxyvitamin D or 1,25-dihydroxyvitamin D or both etc.)

Lines 132-135. For the vitamin D blood serum subgroups, the variable ln{average daily dosage)} is very confusing, as the natural log of 0 is undefined (those with no daily Vit D (thus ln{0}). In lines 99-102, the authors describe weighting the average dose by the numbers of days supplies during the observation period, but this would still not seem to amend an undefined value for those with no vitamin D (the control groups). In which case, how were the controls employed in the subgroup analysis? A better description of this variable and how it was employed in the models would be very helpful.

Results:

Did the authors consider the condition of TBI as a potential covariate, give that in this veteran population TBI may be frequent, may differ between exposure groups, and could impact the outcomes of interest?

Do the authors have access to vitamin D blood levels after the onset of supplementation? As if so, analysis by those that reached sufficient levels, versus those with continued insufficiency and/or deficiency may be interesting.

Discussion:

Sentence starting “ A secondary analysis of these data by suggested…….” A word is missing here.

Final paragraph: Reference should be included in this paragraph starting from the sentence “Other trials included subjects with…..”.

Conclusion: Second line of paragraph- it would be better for the authors refer to the veterans by another definition, rather than “Patients”.

Table 1: There are some mistakes in the tables: (e.g. Table 1 age of the D3 post-matched controls; Table 2, 0-19 D2 cohort numbers of control N and treated N, do no sum to the total patient N). Checking of the tables is advised."

Reviewers' comments:

Reviewer's Responses to Questions

**Comments to the Author**

1. Is the manuscript technically sound, and do the data support the conclusions?

Reviewer #1: Yes

Reviewer #2: Yes

2. Has the statistical analysis been performed appropriately and rigorously? 

Reviewer #1: I Don't Know

Reviewer #2: Yes

3. Have the authors made all data underlying the findings in their manuscript fully available?

Reviewer #1: No

Reviewer #2: Yes

4. Is the manuscript presented in an intelligible fashion and written in standard English?

Reviewer #1: Yes

Reviewer #2: Yes

5. Review Comments to the Author

Reviewer #1: I found this study very interesting and very revealing. I was very enthusiastic about Vitamin D 10-15 years ago and did a small study in people with schizophrenia but became disenchanted with it after a succession of negative studies were published, mainly in depression. I was very glad to see the review of these studies in this paper revealing their inadequacies and emphasizing that suicidal ideation and behavior were not assessed. But I wonder if that review might be more appropriate in the introduction rather than the discussion.

This study looks at suicide attempts and intentional self-harm resulting in emergency room or in-patient admission, which are much more definite and specific than depression and have some different brain mechanisms. It is a retrospective study using Veterans Affairs and Medicare electronic data, from 2010 to 2018, with patients prescribed Vitamin D matched 1:1 to untreated controls. It uses a very large sample: over half a million treated, nearly two million not treated controls: or is it three and a half million not treated? (D2 Controls plus D3 Controls.) Why are the numbers of Pre-matching D3 controls not the same as Pre-matching D2 controls? If they are a different pool of controls how were the controls chosen for each pool? Some further explanation might be useful. And related: why are there only 9,534 D3 controls with 0-19 Vitamin D level when there are 53,881 D2 controls with that level?

The treated patients were included from the date of their prescription. Were the untreated matched by that starting date?

It is fortuitous (and surprising to me) that so many had Vitamin D blood levels, especially as many with low levels were not treated (see below)

The dose used is mentioned and analyzed using the natural logarithm of average dosage (why?). I think mention of the range of dose used would be useful – especially as there is discussion of the appropriate dose under Practical and Policy Implications – but not based on the results of this study.

I don’t understand Propensity Score matching despite reading several descriptions on line. I will accept that it is legitimate. Someone experienced with it should decide its appropriateness. Is everyone expected to know what it means? If not, some more explanation might help. For example, if I look at Depression can I presume that for each Treated person there is a Control of the same gender, race, age (how close?) and any other characteristic?

Typo Table 1, Post-Matching Restricted Sample Control column, Age ?61 not 6.

Table 1 does confirm that people with Bipolar, Depression and other psychiatric conditions were more likely to be given Vitamin D although a greater number with these conditions were not treated, but the matching brought the numbers and the treatment likelihood to the same. The same is true for Vitamin D deficiency. All good.

Table 2, typo: Suicide Attempt by Control column, 0-19 Vitamin D Level Cohort line, should be 86, not 6. And some of the numbers don’t add up; for example, Vitamin D2 females: if there are 45,277 total females the treated should be about 22,742 not 227,420. It does look as if, when Vitamin D levels were very low, D3 was used preferentially for treatment but still about 63,000 with levels 0-19 were untreated. That raises the question of why blood levels were taken.

The Vitamin D2 Level Cohorts Suicide Attempts are surprising (not much difference) given the big differences in race and gender cohorts. That makes it hard to understand the last sentence of the Results section: “However, models of the serum level subsamples demonstrated notably lower rates with increasing blood serum levels relative to the full sample.” Is that for D2 as well as D3 or just the latter? And is there a higher rate of suicide attempts with Vitamin D when the blood level was over 40. (Yes, the numbers are too small. Still it reminds me of the association between maternal (or cord) blood levels and schizophrenia in the offspring – a J-shaped curve.)

Presumably the mental health comorbidities were taken from any time during the two years of treatment (or control). It is interesting to me that the beneficial effects of treatment continue with little change for the two years (Figures 1 and 2) when blood levels of Vitamin D normalize with treatment in two to three months.

Under Discussion, second paragraph, line 4, typo: did should be died. In the sixth line the word “by” presumably should not be there. I think it would be useful to say exactly what is meant by “precision medicine” – presumably only treating those with low Vitamin D levels.

Overall, I think it is a very good study, good methodology, with important findings.

Reviewer #2: In this interesting study, Lavigne & Gibbons set out to examine the benefits of exposure to Vitamin D, on the outcome of suicide. They performed a retrospective cohort study, using the extensive VA database of medical and pharmacy records, covering a period of approximately 8 years. Cox proportional hazards models were employed to estimate the association between vitamin D exposure (D2 or D3), on suicide attempt or death, with stratified analysis also included to assess the impact of race, gender, dosage and vitamin D blood serum levels.

The rationale behind the study is supported by the growing body of evidence cited in the Introduction, suggesting that low levels of vitamin D are associated with an increased risk of suicidal behavior. The methods are appropriate and the conclusions valid. There are only a few amendments that are suggested to improve clarity of the manuscript as currently written

Methods:

It seems from Table 1 that the pre- and post-matched control groups for Vitamin D2 and D3 were independent of each other (as the starting numbers are different). Statistically speaking, this is a benefit and should be noted in the methods.

Line 86. The authors should specify the vitamin D assays employed for blood measures (e.g. 25-hydroxyvitamin D or 1,25-dihydroxyvitamin D or both etc.)

Lines 132-135. For the vitamin D blood serum subgroups, the variable ln{average daily dosage)} is very confusing, as the natural log of 0 is undefined (those with no daily Vit D (thus ln{0}). In lines 99-102, the authors describe weighting the average dose by the numbers of days supplies during the observation period, but this would still not seem to amend an undefined value for those with no vitamin D (the control groups). In which case, how were the controls employed in the subgroup analysis? A better description of this variable and how it was employed in the models would be very helpful.

Results:

Did the authors consider the condition of TBI as a potential covariate, give that in this veteran population TBI may be frequent, may differ between exposure groups, and could impact the outcomes of interest?

Do the authors have access to vitamin D blood levels after the onset of supplementation? As if so, analysis by those that reached sufficient levels, versus those with continued insufficiency and/or deficiency may be interesting.

Discussion:

Sentence starting “ A secondary analysis of these data by suggested…….” A word is missing here.

Final paragraph: Reference should be included in this paragraph starting from the sentence “Other trials included subjects with…..”.

Conclusion: Second line of paragraph- it would be better for the authors refer to the veterans by another definition, rather than “Patients”.

Table 1: There are some mistakes in the tables: (e.g. Table 1 age of the D3 post-matched controls; Table 2, 0-19 D2 cohort numbers of control N and treated N, do no sum to the total patient N). Checking of the tables is advised.

6. PLOS authors have the option to publish the peer review history of their article (what does this mean?). If published, this will include your full peer review and any attached files.

Reviewer #1: **Yes: **Nigel Bark

Reviewer #2: No

---

## [Author Response · Author response to Decision Letter 0]

7 Nov 2022

Response to Editors

Author Response: We have updated our formatting and file naming conventions to be consistent with journal requirements

2. Please provide additional details regarding participant consent. In the ethics statement in the Methods and online submission information, please ensure that you have specified (1) whether consent was informed and (2) what type you obtained (for instance, written or verbal, and if verbal, how it was documented and witnessed). If your study included minors, state whether you obtained consent from parents or guardians. If the need for consent was waived by the ethics committee, please include this information. If you are reporting a retrospective study of medical records or archived samples, please ensure that you have discussed whether all data were fully anonymized before you accessed them and/or whether the IRB or ethics committee waived the requirement for informed consent. If patients provided informed written consent to have data from their medical records used in research, please include this information.

Author Response: We have added language around the exempt determination status from the VA IRB in the last paragraph of the methods section.

Author Response: We have updated or funding statement to include this sentence. See below.

Funding Statement: This study was supported, in part, with resources from the Department of Veterans Affairs Center of Excellence for Suicide Prevention in support of the 2020 call by the Department of Veterans Affairs for research to support the national response to the COVID-19 pandemic. This study does not represent the views of the Department of Veterans Affairs or the United States Government. Dr. Gibbons had full access to all the data in the study and takes responsibility for the integrity of the data and the accuracy of the data analysis. The funders had no role in study design, data collection and analysis, decision to publish, or preparation of the manuscript.

4. Thank you for stating the following in the Acknowledgments Section of your manuscript. We note that you have provided funding information that is not currently declared in your Funding Statement. However, funding information should not appear in the Acknowledgments section or other areas of your manuscript. We will only publish funding information present in the Funding Statement section of the online submission form. Please remove any funding-related text from the manuscript and let us know how you would like to update your Funding Statement. Please include your amended statements within your cover letter; we will change the online submission form on your behalf.

Author Response: See previous response for our funding statement. We have removed it from the text of our manuscript.

5. In your Data Availability statement, you have not specified where the minimal data set underlying the results described in your manuscript can be found. PLOS defines a study's minimal data set as the underlying data used to reach the conclusions drawn in the manuscript and any additional data required to replicate the reported study findings in their entirety. All PLOS journals require that the minimal data set be made fully available. For more information about our data policy, please see http://journals.plos.org/plosone/s/data-availability."Upon re-submitting your revised manuscript, please upload your study’s minimal underlying data set as either Supporting Information files or to a stable, public repository and include the relevant URLs, DOIs, or accession numbers within your revised cover letter. For a list of acceptable repositories, please see http://journals.plos.org/plosone/s/data-availability#loc-recommended-repositories. Any potentially identifying patient information must be fully anonymized. Important: If there are ethical or legal restrictions to sharing your data publicly, please explain these restrictions in detail. Please see our guidelines for more information on what we consider unacceptable restrictions to publicly sharing data: http://journals.plos.org/plosone/s/data-availability#loc-unacceptable-data-access-restrictions. Note that it is not acceptable for the authors to be the sole named individuals responsible for ensuring data access. We will update your Data Availability statement to reflect the information you provide in your cover letter.

Author Response: See below for our data availability statement.

Data availability statement: The data that support the findings of this study are available from the United States Department of Veterans Affairs, but restrictions apply to the availability of these data, which were used for the current study, and so are not publicly available. Data are, however, available from the authors upon reasonable request and with permission of the United States Department of Veterans Affairs.

6. Please ensure that you include a title page within your main document. You should list all authors and all affiliations as per our author instructions and clearly indicate the corresponding author.

Author Response: We have updated the corresponding information on the title page.

Author Response: We have updated our ethics statement in the last paragraph of the methods section of the manuscript. See below:

“This study (VA MIRB # 00701, PI Jill Lavigne) was reviewed and approved under Category 4 exempt determination by the Syracuse VA Medical Center Institutional Review Board in Syracuse, New York, and the VA’s VIREC Office for Medicare data. This research did not meet the criteria for humans subjects research because the VA Corporate Warehouse Data and Medicare claims are de-identified, so informed consent was not required. All methods were performed in accordance with the relevant guidelines and regulations.”

Reviewer 1 Comments & Author Responses

I was very glad to see the review of these studies in this paper revealing their inadequacies and emphasizing that suicidal ideation and behavior were not assessed. But I wonder if that review might be more appropriate in the introduction rather than the discussion.

Author Response: We now review these studies in the third paragraph of the introduction section and discuss them further in the discussion section. The extension of this discussion is done to further contextualize our research findings in light of the previous literature. 

This study looks at suicide attempts and intentional self-harm resulting in emergency room or in-patient admission, which are much more definite and specific than depression and have some different brain mechanisms. It is a retrospective study using Veterans Affairs and Medicare electronic data, from 2010 to 2018, with patients prescribed Vitamin D matched 1:1 to untreated controls. It uses a very large sample: over half a million treated, nearly two million not treated controls: or is it three and a half million not treated? (D2 Controls plus D3 Controls.) 

Author Response: To be clear, before matching, there were 1,764,309 possible controls in the D2 sample and 1,907,720 in the D3 control sample. Note that there is substantial overlap in these control groups, and the total number of unique controls is closer to 2 million. 

Why are the numbers of Pre-matching D3 controls not the same as Pre-matching D2 controls? If they are a different pool of controls how were the controls chosen for each pool? Some further explanation might be useful.

Author Response: The number of controls for each supplement is different because D2 controls were dropped if they had taken D3 at any point in the study period, and D3 controls can’t have taken D2 at any point in the study period to clarify this point. We have added these details to the description in the study population section of the methods section:

“We also excluded patients who had one or more prescriptions for vitamin D2 between 2010-2020 from the vitamin D3 control group and patients who had one or more prescriptions for vitamin D3 between 2010-2020 from the vitamin D2 control group. This was done to prevent Vitamin D3 and D2 treated patients from being used as controls for the D2 and D3 analyses, respectively.”

And related: why are there only 9,534 D3 controls with 0-19 Vitamin D level when there are 53,881 D2 controls with that level?

Author Response: There was a typo during the construction of the table. The correct number for D2 controls is 3,800 and D2 treated is 53,881. The ratio of D2 untreated to treated at 0-19 ng/ml is similar to D3 after this correction. See table 2.

The treated patients were included from the date of their prescription. Were the untreated matched by that starting date?

Author Response: Matching was only done on included covariates. After matching, treated and control veterans were followed from the date of the treated pair’s index prescription, so yes, they were matched in terms of the index date in terms of the analysis of suicidal events. See the statistical analysis subsection of the methods section for clarification.

The dose used is mentioned and analyzed using the natural logarithm of average dosage (why?).

Author Response: This was done to give the parameter estimate interpretations of a percentage point change in dosage. Our motivation for this is now clarified in the exposure subsection of the methods section.

“Single unit increases in vitamin D dosage (i.e., 1 ng/ml) are unlikely to be related to changes in suicide and self-harm risk, which motivated our use of the natural logarithm of the average daily dosage.”

I think mention of the range of dose used would be useful – especially as there is discussion of the appropriate dose under Practical and Policy Implications – but not based on the results of this study.

Author Response: we have added the range of vitamin D dosages observed in the data in the first paragraph of the exposure section.

“The average vitamin D dosage was obtained by taking the average of the prescribed or administered dosages (which ranged from 40 UI to 50,000 UI) weighted by days supplied during the observation period during which the patient filled prescriptions for vitamin D (i.e., two years following index prescription).”

I don’t understand Propensity Score matching despite reading several descriptions online. I will accept that it is legitimate. Someone experienced with it should decide its appropriateness. Is everyone expected to know what it means? If not, some more explanation might help. For example, if I look at Depression can I presume that for each Treated person there is a Control of the same gender, race, age (how close?) and any other characteristic?

Author Response: We see in a comment below you resolved some of this confusion over the use of propensity score matching. Either way, we have added some additional language around the motivation in the last sentence of the first paragraph of the study population subsection of the methods section.

“Propensity score matching was performed to help balance the treated and control populations across characteristics known to be associated with both vitamin D supplementation and suicidal and self-harm behaviors.”

Typo Table 1, Post-Matching Restricted Sample Control column, Age ?61 not 6.

Author Response: We have fixed this typo. See table 1.

Table 2, typo: Suicide Attempt by Control column, 0-19 Vitamin D Level Cohort line, should be 86, not 6. And some of the numbers don’t add up; for example, Vitamin D2 females: if there are 45,277 total females the treated should be about 22,742 not 227,420. 

Author Response: The value of 6 is correct, but the number of controls with 0-19 ng/ml in the vitamin d2 population should be 3,800 instead of 53,881. We have fixed this issue and some other data entry typos in this table. See updated table 2.

It does look as if, when Vitamin D levels were very low, D3 was used preferentially for treatment but still about 63,000 with levels 0-19 were untreated. That raises the question of why blood levels were taken.

Author Response: D3 is generally more common than D2. It is also possible that some portion of the 63,000 that received a test were asked/decided to take vitamin D over the counter. This is mentioned as a possible limitation of the study but likely makes our results more conservative (as the controls may have been receiving treatment).

The Vitamin D2 Level Cohorts Suicide Attempts are surprising (not much difference) given the big differences in race and gender cohorts. That makes it hard to understand the last sentence of the Results section: “However, models of the serum level subsamples demonstrated notably lower rates with increasing blood serum levels relative to the full sample.” Is that for D2 as well as D3 or just the latter? And is there a higher rate of suicide attempts with Vitamin D when the blood level was over 40. (Yes, the numbers are too small. Still it reminds me of the association between maternal (or cord) blood levels and schizophrenia in the offspring – a J-shaped curve.)

Author Response: We have rewritten the sentence for additional clarity. It is curious that those with testing had lower attempt rates than those that did not have a test, which we have called attention to. Our secondary point, which is now at the end of the sentence, is that the attempt and self-harm rates were generally lower in the control groups at higher blood levels.

“However, the serum level subsamples in both the Vitamin D2 and Vitamin D3 demonstrated notably lower attempt rates than the full sample. It may therefore be the case that those who have their vitamin D levels tested are characteristically different from those that did not receive testing. Finally, as vitamin D levels increased, the proportion of veterans experiencing suicide attempts and self-harm declined in the control groups for Vitamin D2 and Vitamin D3 and stayed relatively fixed in the treated groups.”

Presumably, the mental health comorbidities were taken from any time during the two years of treatment (or control). 

Author Response: We have clarified that comorbidities span all possible medical claims for each veteran in the second to last sentence of the study population subsection of the methods section.

“Health conditions were identified using the presence of one or more ICD-9 or ICD-10 diagnoses on any medical claim for a veteran between 2010-2018.”

It is interesting to me that the beneficial effects of treatment continue with little change for the two years (Figures 1 and 2) when blood levels of Vitamin D normalize with treatment in two to three months.

Author Response: This is a good observation. The risk appears to be higher while blood levels are being established but continues to decrease over time, likely because those who were at high risk of an attempt have already been censored because they made an attempt (the same effect is observed in the controls but to a lesser degree). The intervention might also continue to be effective over time, even after levels have become more normal.

Under Discussion, second paragraph, line 4, typo: did should be died. 

Author Response: We have fixed this typo

In the sixth line the word “by” presumably should not be there. 

Author Response: We have fixed this typo

I think it would be useful to say exactly what is meant by “precision medicine” – presumably only treating those with low Vitamin D levels.

Author Response: We have added some context to this sentence as you suggest.

“This need for precision medicine (i.e., focusing Vitamin D supplementation on those with particularly low levels) approaches is further supported by twelve clinical trials of vitamin D supplementation and depression reported between January 2010 and May 2020 with overall negative results”

Reviewer 2 Comments & Author Responses

"Methods: It seems from Table 1 that the pre- and post-matched control groups for Vitamin D2 and D3 were independent of each other (as the starting numbers are different). Statistically speaking, this is a benefit and should be noted in the methods.

Author Response: To be clear, before matching, there were 1,764,309 possible controls in the D2 sample and 1,907,720 in the D3 control sample. Note that there is substantial overlap in these control groups, and the total number of unique controls is closer to 2 million. The number of controls for each supplement is different because D2 controls were dropped if they had taken D3 at any point in the study period, and D3 controls can’t have taken D2 at any point in the study period to clarify this point. We have added more description in the study population section of the methods section:

“We also excluded patients who had one or more prescriptions for vitamin D2 between 2010-2020 from the vitamin D3 control group and patients who had one or more prescriptions for vitamin D3 between 2010-2020 from the vitamin D2 control group. This was done to prevent Vitamin D3 and D2 treated patients from being used as controls for the D2 and D3 analyses, respectively.”

Line 86. The authors should specify the vitamin D assays employed for blood measures (e.g. 25-hydroxyvitamin D or 1,25-dihydroxyvitamin D or both etc.)

Author Response: We have specified that this includes both. In the second paragraph of the study population subsection of the methods section.

“Vitamin D blood serum levels were identified by the last 25-hydroxyvitamin D or 1,25-dihydroxyvitamin D test prior to the index prescription of a vitamin D supplement for a patient in either the D2 or D3 treatment group or the corresponding treated pair’s index prescription for the untreated controls.”

Lines 132-135. For the vitamin D blood serum subgroups, the variable ln{average daily dosage)} is very confusing, as the natural log of 0 is undefined (those with no daily Vit D (thus ln{0}). In lines 99-102, the authors describe weighting the average dose by the number of days supplies during the observation period, but this would still not seem to amend an undefined value for those with no vitamin D (the control groups). In which case, how were the controls employed in the subgroup analysis? A better description of this variable and how it was employed in the models would be very helpful.

Author Response: We have forced these values to be 0 manually. We have clarified this in in the first paragraph of the Exposure subsection of the Methods section. 

“For patients not taking vitamin D, we set the average daily dosage and its natural logarithm to be 0 to prevent constructing undefined values..”

Results:

Did the authors consider the condition of TBI as a potential covariate, given that in this veteran population TBI may be frequent, may differ between exposure groups, and could impact the outcomes of interest?

Author Response: We did not control for TBI in our analyses. This is mentioned as a potential limitation in the limitations subsection of the discussion section.

“Further, although we attempt to control for some characteristics likely to be associated with vitamin D supplementation and/or suicidal and self-harm behaviors, there are many unobservable characteristics (for example, traumatic brain injury common among veterans) that could confound our results.”

Do the authors have access to vitamin D blood levels after the onset of supplementation? 

As if so, analysis by those that reached sufficient levels, versus those with continued insufficiency and/or deficiency may be interesting.

Author Response: This is a potentially interesting extension of our work, unfortunately, our two-year follow-up period makes it difficult to study this, given the infrequent number of patients receiving regular lab tests. We have called for a prospective study where blood levels are routinely obtained to in the last sentence of the first paragraph of the discussion section. 

“However, additional research is required to determine what happens to suicidal and self-harm behavior risk once sufficient levels of vitamin D are obtained and is left for future research.”

Discussion:

Sentence starting “A secondary analysis of these data by suggested…….” A word is missing here.

Author Response: We have revised this sentence. 

“A secondary analysis of these data suggested the need for further research on vitamin D, including precision medicine approaches specifically for suicide prevention.34”

Final paragraph: Reference should be included in this paragraph starting from the sentence “Other trials included subjects with”.

Author Response: We have added some references to this sentence.

“Other trials included subjects with relatively normal or high 25(OH)D levels.14-21“

Conclusion: The second line of the paragraph- it would be better for the authors to refer to the veterans by another definition, rather than “Patients”.

Author Response: We have revised this sentence. We have also replaced all instances of the word “patients” with “veterans” throughout the manuscript.

“Veterans receiving higher prescription daily dosages were associated with lower risk or suicide attempt than veterans receiving lower dosages, and the associated risk reduction was more significant among Black veterans, among whom low Vitamin D serum levels are more common than among White veterans.”

Table 1: There are some mistakes in the tables: (e.g., Table 1 age of the D3 post-matched controls; Table 2, 0-19 D2 cohort numbers of control N and treated N, do no sum to the total patient N). Checking of the tables is advised."

Author Response: We have reviewed all tables and updated them with the correct numbers. Thank you for flagging these typos.

---

## [Decision Letter · Decision Letter 1]

1 Dec 2022

The Association between Vitamin D Serum Levels, Supplementation, and Suicide Attempts and Intentional Self-Harm

PONE-D-22-20954R1

Dear Dr. Gibbons,

We’re pleased to inform you that your manuscript has been judged scientifically suitable for publication and will be formally accepted for publication once it meets all outstanding technical requirements.

Kind regards,

James D. Clelland, Ph.D.

Academic Editor

PLOS ONE

Additional Editor Comments (optional):

Reviewers' comments:

Reviewer's Responses to Questions

**Comments to the Author**

1. If the authors have adequately addressed your comments raised in a previous round of review and you feel that this manuscript is now acceptable for publication, you may indicate that here to bypass the “Comments to the Author” section, enter your conflict of interest statement in the “Confidential to Editor” section, and submit your "Accept" recommendation.

Reviewer #1: All comments have been addressed

Reviewer #2: All comments have been addressed

2. Is the manuscript technically sound, and do the data support the conclusions?

Reviewer #1: Yes

Reviewer #2: Yes

3. Has the statistical analysis been performed appropriately and rigorously? 

Reviewer #1: Yes

Reviewer #2: Yes

4. Have the authors made all data underlying the findings in their manuscript fully available?

Reviewer #1: Yes

Reviewer #2: Yes

5. Is the manuscript presented in an intelligible fashion and written in standard English?

Reviewer #1: Yes

Reviewer #2: Yes

6. Review Comments to the Author

Reviewer #1: I appreciate the authors' responses to my comments and alterations or corrections where appropriate. All were addressed.

Reviewer #2: The authors have been very responsive to the previous reviewers comments. This manuscript is now ready to be published.

7. PLOS authors have the option to publish the peer review history of their article (what does this mean?). If published, this will include your full peer review and any attached files.

Reviewer #1: **Yes: **Nigel Bark

Reviewer #2: No

---

## [Editor Report · Acceptance letter]

5 Jan 2023

PONE-D-22-20954R1 

The Association between Vitamin D Serum Levels, Supplementation, and Suicide Attempts and Intentional Self-Harm 

Dear Dr. Gibbons:

I'm pleased to inform you that your manuscript has been deemed suitable for publication in PLOS ONE. Congratulations! Your manuscript is now with our production department. 

Kind regards, 

on behalf of

Dr. James D. Clelland 

Academic Editor

PLOS ONE